# The Association between Cesarean Section Delivery and Child Behavior: Is It Mediated by Maternal Post-Traumatic Stress Disorder and Maternal Postpartum Depression?

**DOI:** 10.3390/bs14010061

**Published:** 2024-01-17

**Authors:** Marie-Andrée Grisbrook, Deborah Dewey, Colleen Cuthbert, Sheila McDonald, Henry Ntanda, Nicole Letourneau

**Affiliations:** 1Faculty of Nursing, University of Calgary, Calgary, AB T2N 4N1, Canada; marie.grisbrook@ucalgary.ca (M.-A.G.); cacuthbe@ucalgary.ca (C.C.); 2Alberta Children’s Hospital Research Institute, Owerko Centre, Calgary, AB T2N 1N4, Canada; dmdewey@ucalgary.ca (D.D.); henry.ntanda@ucalgary.ca (H.N.); 3Department of Community Health Sciences, Cumming School of Medicine, University of Calgary, Calgary, AB T2N 4N1, Canada; sheila.mcdonald@albertahealthservices.ca; 4Department of Pediatrics, Cumming School of Medicine, University of Calgary, Calgary, AB T2N 4N1, Canada; 5Hotchkiss Brain Institute, Cumming School of Medicine, University of Calgary, Calgary, AB T2N 4N1, Canada; 6Department of Oncology, Cumming School of Medicine, University of Calgary, Calgary, AB T2N 4N1, Canada; 7Department of Psychiatry, Cumming School of Medicine, University of Calgary, Calgary, AB T2N 4N1, Canada

**Keywords:** cesarean section, postpartum depression, post-traumatic stress disorder, child behavior, maternal mental health, conditional process modeling

## Abstract

Cesarean sections (C-sections) account for up to 21% of births worldwide. Studies have linked delivery via C-section with an increased risk of child behavior problems, such as internalizing and externalizing behaviors. Maternal postpartum depression (PPD) is also linked to child behavioral problems and may play a mediating role in the association between the mode of delivery and child behavior. Mixed findings between mode of delivery and PPD may be due to a failure to distinguish between C-section types, as unplanned/emergency C-sections are linked to post-traumatic stress disorder (PTSD), which has been linked to PPD. The objectives of this study were to determine whether, (1) compared with spontaneous vaginal delivery (SVD) and planned C-section, unplanned/emergency C-sections are associated with increased child behavior problems at two to three years of age and (2) maternal PTSD and PPD mediate the association between delivery type and child behavior problems. A secondary data analysis was conducted on 938 mother–child dyads enrolled in the Alberta Pregnancy Outcomes and Nutrition (APrON) study. Conditional process modeling was employed. Child behavior was assessed using the Child Behavior Checklist (CBCL) 1.5–5 years, and maternal PPD and PTSD were assessed using the Edinburgh Postnatal Depression Scale (EPDS) and the Psychiatric Diagnostic Screening Questionnaire (PDSQ), respectively. No associations were found between delivery type and child behaviors; however, the indirect effect of emergency C-section on child behaviors was significant via the mediating pathway of maternal PTSD on PPD symptoms.

## 1. Introduction

Cesarean section (C-section) is a delivery method that has seen its worldwide prevalence rate increase from 7% to 21% in the last 30 years, with rates as high as 43% in Latin America and the Caribbean [1]. In North America, C-section delivery accounts for 32% of all deliveries [1]. A C-section can be the safest way to deliver in medically indicated situations, such as labor dystocia, abnormal fetal heart rate, fetal malpresentation, and multiple gestation [2]. However, birth via C-section has also been linked to altered long-term pediatric mental health outcomes, with over 10% of children born via C-section demonstrating internalizing (overcontrolled, e.g., anxiety, depressiveness, withdrawn) and externalizing (undercontrolled, e.g., aggression, hyperactivity) behavioral problems [3,4]. Research evaluating the association between mode of delivery and child behavioral outcomes is limited; it has been hypothesized that maternal mental health, such as postpartum depression (PPD) and post-traumatic stress disorder (PTSD), may be important mediators of this association [5].

PPD has a prevalence rate of approximately 14% in first-year postpartum [6], and mixed findings link the mode of delivery to PPD [7,8,9]. The variability in findings for an association between delivery type and PPD may be due to previous studies’ failure to distinguish between the type of C-section (planned vs. unplanned/emergency) or perhaps a lack of consideration for the role of postpartum PTSD. Mothers’ perceptions of a more negative childbirth experience, such as unplanned/emergency C-sections, are linked to PTSD [10], which is related to PPD [11]. Research has shown that PTSD is a mediating factor in the association between C-section type (planned vs. unplanned) and PPD [12]. With the high rates of C-section deliveries worldwide and their link with childhood internalizing and externalizing behaviors, understanding the pathways through which the association between mode of delivery and child behavioral problems occurs is essential to help guide interventions that can address the impact on families. 

### 1.1. Background

#### 1.1.1. Child Behavior

Globally, child behavioral problems have a 10 to 14% prevalence rate among children aged 18 to 30 months [3,4]. Behavioral problems in childhood are conceptualized as internalizing and externalizing. Internalizing behaviors include anxiety, depression, phobias, and separation anxiety symptoms, and externalizing problems include hyperactivity, aggression, and oppositional defiance symptoms [5]. Internalizing and externalizing problems in childhood predict altered school adaptation [13] and reduced school achievement into adolescence [14]. Behavioral problems are also associated with increased parenting stress [15]. They also put children at risk of altered mental health (mood disorders) into adolescence and adulthood. Ten to twenty percent of children and youth, aged three to seventeen years of age, experience mental health disorders [16,17]. Onset typically occurs in childhood or adolescence, with approximately half of all lifetime mental health disorders starting in adolescence [18]. The rate of pediatric mental health emergency visits has increased in Canada (75%) and the United States (45%) [19,20]. The increasing prevalence of mental health disorders and their long-term impact on children and families support the need for research on factors contributing to early childhood behavioral problems.

#### 1.1.2. C-Section and Child Behavior

Findings of an association between mode of delivery, specifically C-section, and child behavioral problems are mixed. A secondary data analysis of a longitudinal birth cohort (*n* = 258) reported that emergency C-section, combined with psychosocial factors such as male sex, younger maternal age, and lower maternal self-efficacy, significantly predicted (R^2^ = 0.84, *p* < 0.001) more externalizing behavioral problems on the Child Behavior Checklist (CBCL) in the first 30 months of life [21]. A longitudinal birth cohort (*n* = 172) of mother–infant dyads noted that children born via C-section had higher mean scores on internalizing (12.05, *p* < 0.05) and total problems (48.07, *p* < 0.05) compared with the internalizing (9.17) and total problems (39.23) scores of children born vaginally without medications, as measured by the CBCL [4]. However, this study did not differentiate between C-section types. A study of preschoolers (*n* = 8900) identified a relationship between C-sections and child behavior, showing that both elective and emergency C-sections were significantly associated with total difficulty scores in the abnormal range on the Strength and Difficulties Questionnaire (SDQ) [3].

Research findings have also revealed no association between C-section delivery and child behavioral problems and a decrease in child behavior problems among children born via C-section. Li, Ye [22], in a large retrospective cohort study (*n* = 4190), determined that behavioral problems among children aged four to six years were the lowest among women who had a C-section on maternal request. Notably, delivery via an emergency C-section, C-sections occurring before the onset of labor, and mothers with antepartum complications were excluded from the study [22]. A large longitudinal cohort study (*n* = 11,134) of a nationally representative sample in Ireland identified no association between emergency C-section or instrumental vaginal delivery and abnormal scores on the SDQ at three years of age [23]. The role of emergency C-sections in the relationship between mode of delivery and child behavioral problems was not considered in many of the studies discussed. This could influence the potential of identifying a direct or indirect relationship between delivery type and child behavioral problems.

#### 1.1.3. C-Section as Predictor of Maternal PPD and PTSD

Delivery via C-section has been linked to dissatisfaction, failure, and reduced self-esteem, which may increase the risk of depression [24]. C-section delivery has been associated with the development of PPD symptoms and psychological stress [8,9,25,26,27,28]. An emergency C-section has been identified as the most significant factor associated with a negative evaluation of the birth experience at one year postpartum [29]. A systematic review of 15 studies with sample sizes varying from *n* = 44 to *n* = 5333 reported that 11 studies found significant associations between women’s perspective of their birth experience and PPD symptoms [30]. The lack of control experienced during emergency obstetrical interventions, such as an emergency C-section, may be associated with the development of depression [31,32] and birth trauma [33] among new mothers. The negative perceptions of birth accompanying emergency obstetrical interventions may contribute to altered postpartum maternal mental health, and subsequently, children’s behavioral problems.

PPD is highly associated with PTSD in the postpartum period. For women with postpartum PTSD over half also have comorbid PPD [34]. The Listening to Mothers II and Listening to Mothers II Postpartum, a two-stage United States national survey, identified that PTSD has a 9% prevalence rate and is associated with obstetrical interventions and PPD symptoms [35]. This association is further supported by a study of 354 mother–child dyads, which identified that mothers who experienced an emergency C-section had increased PTSD scores, as measured by the Psychiatric Diagnostic Screening Questionnaire (PDSQ), which was associated with PPD scores on the Edinburgh Postnatal Depression Scale (EPDS) [12]. Poor maternal emotional state in the postpartum period is a risk factor for emotional and behavioral problems among children [4]. The link between postpartum PTSD and PPD may affect how birth experiences impact child behavior. 

#### 1.1.4. Other Predictors of Children’s Behavioral Problems

A secondary analysis of a longitudinal birth cohort study determined that in addition to C-section delivery, demographic factors such as maternal education, maternal age, and child sex predicted child behavior problems [21]. Maternal age and education were linked to increased internalizing problems in children, such as emotional reactivity, anxious/depressed/depressiveness, somatic complaints, and withdrawal; in addition, boys were found to display more internalizing, externalizing, and total behavioral problems than girls [4]. Maternal depression was also found to interact with child sex; prenatal and postnatal depression were risk factors for behavioral symptoms in girls and boys [36]. 

This study aimed to provide a more in-depth understanding of whether the mode of delivery was associated with child behavioral problems via pathways of PTSD and PPD. The study addressed the following research questions: (1) Is unplanned/emergency C-section, compared with planned C-section or spontaneous vaginal delivery (SVD), associated with more child behavioral problems at two to three years of age? (2) Is the association between the mode of delivery and children’s behavioral problems at two to three years of age mediated by maternal PTSD and PPD? We hypothesized that compared with an SVD or planned C-section, an unplanned/emergency C-section may not be directly associated with more child behavioral problems at two to three years of age. However, we hypothesized that the association between mode of delivery and children’s behavioral problems at two to three years of age would be mediated by maternal PTSD and maternal PPD (Figure 1). 

## 2. Materials and Methods

### 2.1. Study Design and Sample

Data from this study were drawn from the Alberta Pregnancy Outcomes and Nutrition (APrON) longitudinal pregnancy cohort study. The APrON study was initially designed to collect data on maternal nutrition and maternal mental health, birth and obstetric outcomes, and child neurodevelopment [37,38]. Participants were recruited from a community sample of women living in two large cities in Alberta, Canada, between 2009 and 2012 [37]. Women over 16 years old and with a gestational age of fewer than 27 weeks at enrollment were eligible. Those excluded were women planning to move away from the region and those unable to complete the questionnaires in English. The sample was constrained to 938 maternal–child dyads born via C-section or SVD. The inclusion criteria were singleton delivery, with complete data on study measures (i.e., EPDS, PDSQ, and CBCL). Informed consent was obtained from participants prior to study participation. The University of Calgary Health Research Ethics Board and the University of Alberta Health Research Ethics Biomedical Panel approved this study (REB14-1702_REN7).

### 2.2. Measures

#### 2.2.1. Predictor: Delivery Type

Medical labor and delivery records were reviewed to determine the mode of delivery. C-sections identified as elective on the delivery record were classified as planned C-sections, with all other C-sections classified as an emergency. The following variables were created: an emergency C-section was coded as 1, and SVD or a planned C-section were coded as 0.

#### 2.2.2. Outcome: Child Internalizing and Externalizing

Child internalizing and externalizing behaviors were measured using mean CBCL scores obtained between two and three years of age. The CBCL for children ages 1.5 to 5 is a 100-item parent/caregiver report questionnaire that assesses parents’ perceptions of their child’s behavior [39]. The responses range from 0 to 2, with 0 indicating “not true”, 1 indicating “somewhat or sometimes true”, and two indicating “very true or often true”. The overall score range is from 0–200. The CBCL assesses domains of internalizing symptoms (emotionally reactive, anxious/depressed, somatic complaints, and withdrawn) and externalizing symptoms (aggression, attention issues) [39]. For the internalizing, externalizing, and total problems, T scores (mean = 50, SD = 10) were calculated. Higher scores indicate more significant problem behaviors [39]. The CBCL can be completed within 20 to 30 min and has high internal consistency (Cronbach alphas of 0.87 and 0.89 for the internalizing and externalizing problem scales, respectively) [40].

#### 2.2.3. Mediator: Maternal Postpartum Depressive Symptoms and Maternal PTSD

The EPDS, used to measure postpartum depressive symptoms, is a 10-question Likert self-report scale with responses ranging from 0 to 3 according to the severity of the symptom [41]. It has been extensively used by clinicians and researchers worldwide to screen for postnatal depression [42]. The overall score can range from 0 to 30. A score of ≥12 represents women likely suffering from depression [41]. The EPDS tool is also validated for use with pregnant women with 100% sensitivity using a cut-off of 12 [43]. The EPDS has been found to have a standardized alpha coefficient of 0.87 and split-half reliability of 0.88 [41]. It is sensitive to changes in the severity of depression over time [41]. The EPDS can be completed in five minutes. Maternal postpartum depressive symptoms were measured at three months postpartum.

Maternal PTSD was measured using the PDSQ at three months postpartum. The PDSQ is a self-report questionnaire, which includes a 15-item subscale evaluating PTSD symptoms, with items scored 0 and 1 for “no” and “yes”, respectively. It has been used in prenatal [44] and postnatal screening [45]. A cutoff score of five represents the presence of PTSD symptoms [46]. The first two questions on the PTSD subscale refer to having ever experienced or witnessed a traumatic event, with the remainder of the questions referring to symptoms experienced in the two weeks before the evaluation [46]. The subscale evaluating PTSD symptoms has a Cronbach alpha of 0.94 and 92% sensitivity.

#### 2.2.4. Covariates

Covariate data for socioeconomic status (education and income) and antenatal depression were gathered at the first study visit and recollected at each assessment point. Labor and delivery records were reviewed for data on the child’s biological sex. Social support was measured at the time of study intake and the second trimester (highest value used in the analyses) using the Social Support Questionnaire [47], a four-item measure with scores ranging from 0 to 16, with higher values indicating greater perceived social support.

### 2.3. Data Analysis

The statistical analyses were performed using Statistical Analysis Systems (SAS) version 9.4 for Windows. Descriptive statistics (*n*, mean, standard deviation, minimum, and maximum) were calculated for continuous variables. Frequencies and percentages were presented for categorical and ordinal variables. 

To examine whether maternal PTSD and maternal PPD mediated the association between the type of delivery and child behavior, a linear-regression-based serial multiple mediation analysis was conducted using the Hayes process SAS macro version 4.3 [48]. Before running the analyses, the data were evaluated for normality, linearity, homoscedasticity, independence, and multicollinearity. The model aimed to investigate total and direct effects (a_1_, a_2_, b_1_, b_2_, d_21_, and c′) as the standardized regression coefficient shows among the dependent and independent variables. The indirect effects (a = a_1_ × b_1_; b = a_2_ × b_2_; c_1_ = a_1_ × d_21_ × b_2_) were examined to determine if a change in child behavior for mothers with unplanned C-section is mediated by maternal PTSD, maternal PPD, or both. All mediation hypotheses were tested using the bootstrapping method, with a resampling procedure of 20,000 to calculate 95% confidence intervals (CIs) [49]. All indirect effects are significant when the *p*-value is <0.05 and the confidence interval does not contain zero. All models were fitted using a random seed for reproducibility of the results.

## 3. Results

### 3.1. Descriptive Findings

Child and maternal demographic data and descriptive characteristics are presented in Table 1. Of the 938 mother–child dyads who participated in the study, 19% delivered via emergency C-section. The mean age of the mothers was 32 years, and most mothers were married (97%), had a household income > CAD 70,000 (84%), and had postsecondary education (76%). Just over half of the children were male (52%). Among the 938 mothers, 6.1% presented with PPD scores ≥ 12, and 3.5% presented with PTSD scores ≥ 5, with 1.2% presenting with both elevated PPD and PTSD scores.

### 3.2. Internalizing Behaviors

The serial multiple mediation model analysis illustrated in Figure 2 below evaluated the association between the mode of delivery and child internalizing behavior. In this analysis, delivery type is associated with maternal postpartum PTSD (a_1_ = 0.253, *p* = 0.042) but not maternal PPD (a_2_ = 0.206, *p* = 0.512). Maternal PTSD is positively associated with maternal PPD (d_21_ = 0.682, *p* < 0.0001) but not with child internalizing behavior (b_1_ = 0.101, *p* = 0.588). Maternal PPD is positively associated with child internalization behavior (b_2_ = 0.419, *p* < 0.0001). Delivery type is not directly associated with child internalizing behavior (c′ = 0.835, *p* = 0.225). The direct effect of delivery type on internalizing behavior remained nonsignificant, even after adjusting for the covariates of prenatal depression, child sex, SES, and perinatal support. 

To test the mediation hypotheses, the serial multiple hypotheses were tested for the association between delivery type and child internalizing behavior (Table 2). The total indirect pathway revealed significant paths for the third pathway of delivery type: maternal PTSD to maternal PPD to child internalizing behavior.

### 3.3. Externalizing Behaviors

The serial multiple mediation model analysis is illustrated in Figure 3 below. In this analysis, delivery type is associated with maternal postpartum PTSD (a_1_ = 0.253, *p* = 0.091) but not maternal PPD (a_2_ = 0.206, *p* = 0.515). Maternal PTSD is positively associated with maternal PPD (d_21_ = 0.682, *p* < 0.0001) but not with child externalizing behavior (b_1_ = 0.249, *p* = 0.188) Maternal PPD is positively associated with child externalizing behavior (b_2_ = 0.342, *p* < 0.0001). Delivery type is not directly associated with child externalizing behavior (c′ = 0.763, *p* = 0.253). The adjusted multiple mediation model analysis adjusted for prenatal depression, child sex, SES, and perinatal support. The direct effect of delivery type on externalizing behavior remained nonsignificant, even when adjusting for the covariates. 

To test the mediation hypotheses, the serial multiple hypotheses were tested using the bootstrap approach for the association between delivery type and child externalizing behavior (Table 3). The total indirect pathway revealed significant paths for the third pathway of delivery type: maternal PTSD to maternal PPD to child externalizing behavior.

## 4. Discussion

Our study is the first to examine the association between C-section delivery and child behavioral problems via the mediating factors of maternal postpartum PTSD and PPD symptoms. No direct association was noted between delivery type (i.e., emergency C-section delivery versus planned C-section or SVD) and child internalizing or externalizing behaviors. However, emergency C-section delivery was indirectly associated with child internalizing and externalizing behaviors via maternal PTSD and PPD symptoms. After controlling for SES, antenatal depression, child sex, and social support, the associations remained significant. Mothers who experienced an emergency or unplanned C-section had a 0.24 increase in PTSD scores (mean 0.6 (SD 1.5)) compared with mothers who underwent a planned C-section or SVD. A one-unit increase in PTSD scores, as measured by the PDSQ, predicted a 0.37 unit increase in EPDS PPD scores (mean 4.8 (SD 3.9)). A unit increase in PPD predicted a 0.29 unit increase in child internalizing T score (mean 44.75 (SD 8.34)) behavioral problems. A one-unit increase in PTSD predicted a 0.37 increase in PPD, and a one-unit increase in PPD predicted a 0.20 increase in externalizing T score (mean 45.81 (8.20)) behavioral problems.

Although the increase is modest, when considering scores that may approach clinical significance, such as T scores > 70 or borderline ranges of 65–69 [39], our findings demonstrate that an emergency C-section may impact behavioral scores, making them high enough to be a concern. These findings suggest that an unplanned/emergency cesarean delivery impacts maternal postpartum PTSD symptoms, leading to increased maternal PPD symptoms and, subsequently, more internalizing and externalizing behaviors among young children. These findings indicate that delivery via an emergency C-section is associated with increased PTSD and PPD symptoms at three months postpartum, resulting in increases in internalizing and externalizing behaviors among children at two to three years of age. Evaluation of this complex relationship demonstrates that the most significant impact is noted between emergency C-section and PTSD scores, as an emergency C-section resulted in a 16% increase in PTSD scores. 

Previous research, although limited, identifies emergency C-sections as an influential factor in the development of childhood behavioral problems [3,50]. Emergency C-sections are associated with higher total SDQ and peer problem scores among five-year-old children [50]. The findings of a study performed by Rutayisire, Wu [3] also support this link; however, it identified an association between elective and emergency C-sections and total abnormal SDQ (17–40) scores among children aged three to six. Although some studies support a direct association between emergency C-section and child behavioral problems, the majority of the literature identifies no association between emergency C-section delivery and child behavioral problems [23,51,52,53,54]. This is consistent with our findings of a lack of direct associations between emergency C-sections and child behavior problems. We hypothesized that birth via emergency C-section would not have a direct link to internalizing and externalizing symptoms among preschool-aged children. Instead, we hypothesized that an effect on child behavior would emerge when maternal postpartum mental health factors (PTSD and PPD) were considered, as they are associated with the mode of delivery, specifically emergency C-section. This may explain why most research has failed to identify an association between mode of delivery, specifically emergency C-section, and internalizing and externalizing symptoms among children. 

Understanding the pathway of emergency C-section to child behavioral problems requires an in-depth evaluation of the factors contributing to significance. Our study offers a possible explanation for the link between the mode of delivery and child behavioral problems. The findings link delivery via emergency C-section with an increase in PTSD symptoms; an increase in PTSD symptoms was linked to an increase in PPD symptoms, and an increase in PPD symptoms was associated with an increase in internalizing and externalizing child behaviors as measured by the CBCL at two to three years of age. 

How women deliver their children can profoundly impact the perceptions of birth experiences. Birth-related PTSD occurs due to intense fear, loss of control, helplessness, or horror in response to an actual or perceived threat during the labor and delivery period [55]. As many as 26% of women experience PTSD symptoms [10], and 9% are diagnosed with PTSD in the postpartum period [35]. Physical problems such as a cesarean incision are significantly related to post-traumatic stress scores in the postpartum period, as measured by the Post-Traumatic Stress Disorder Symptom Scale-Self Report (PSS-SR) [35]. Delivery via C-section is the most common birth intervention related to developing birth-related PTSD [56]. As noted, PTSD is highly comorbid with PPD, and there is a shared vulnerability underlying the development of PTSD and PPD [35]. 

The evidence linking maternal PPD to altered child behavior outcomes is well-established [5,57,58,59,60]. Maternal PPD has been linked to more behavioral problems at two years of age, an increased prevalence of mood disorders at 19 months of age, and internalizing problems at one year of age [60], as well as externalizing behaviors at up to 16 years of age [5]. Evidence suggests that perinatal disorders, such as maternal PPD, impact parenting quality as symptoms of mental disorders affect an individual’s ability to respond to their environment and thus their parenting capabilities [5]. Research has also shown that maternal depression in the first year of life is significantly associated with higher scores on internalizing and total problems scales on the CBCL in children [57]. The link between the mode of delivery and child behavioral problems appears to be more complex than mediation via maternal PPD. Our study did not identify an association between emergency C-section, PPD, and child internalizing or externalizing symptoms. However, a significant indirect relationship was noted via mediating factors of both maternal PTSD and PPD and child internalizing and externalizing symptoms. 

This study builds on our prior research evaluating the association between mode of delivery and maternal PPD symptoms that identified no direct relationship between mode of delivery and maternal PPD symptoms [12]. However, an indirect association was noted via PTSD symptoms as a mediator, highlighting the importance of obstetrical complications in perceptions of subjective birth experiences and the impact on maternal postpartum mental health [12]. Women delivering via an emergency C-section are more likely to present with childbirth-related PTSD and be at greater risk of developing PPD. 

Gaining knowledge of the complex factors involved in the relationship between C-section delivery and child behavioral problems could help guide interventions to improve outcomes for families. Maternity care providers should be aware of how delivery factors impact maternal and pediatric mental health. Perinatal healthcare providers should work to enhance factors that are protective against postpartum PTSD and PPD, such as birth preparedness, newborn contact after delivery, health practices, and social support [10,35,60,61]. Maternity care providers are uniquely positioned to support women in developing protective factors, as they can educate women about different birth scenarios, encourage postpartum skin-to-skin contact, and employ caring and effective communication strategies during labor and delivery events. Using a caring and supportive approach and communication are identified as strategies that can help prevent birth trauma [55]. Social support is consistently reported as a protective factor as it has a significant negative relationship with maternal postpartum PTSD [35,62] and PPD [59,60,63]. Additionally, consideration should be given to implementing universal screening of maternal postpartum PPD, as perinatal depression screening is associated with a reduced risk of depression [64]. Assessment of birth-related PTSD should also be considered when risk factors such as an emergency C-section are present. Knowledge of the importance of screening for mental health conditions in the postpartum period and of PTSD and PPD risk factors and protective factors will enable perinatal care providers to incorporate protective factors into their practice, identify women at risk, and reduce the risk of childhood behavior problems. 

## 5. Strengths and Limitations

The use of a multiple linear regression model allowed us to examine the plausibility that maternal mental health is the mechanism through which C-section affects child behavior [48]. Our study did not aim to identify causation between C-section and child behavior, but rather, the study aimed to understand how C-section delivery exerts its effect on child behavior. The prospective longitudinal design of this study is a strength as it allowed for the follow-up of participants from pregnancy through to three years postpartum. The study used validated tools for measurement of the outcome variables such as PTSD, PPD, and child behavior. The sample size allowed the analysis to control for relevant covariates while maintaining adequate power to support the findings. Although the initial data were collected over 12 years ago, the prevalence rates for C-section delivery, PTSD, and PPD have remained stable. This study used three-month data for maternal PPD and PTSD as the postpartum onset of PPD is typically diagnosed in the four weeks following delivery [65], with the highest onset of PPD and the greatest risk of mental health disorder admissions occurring in the first three months following delivery [9,65,66]. Given that maternal depression extending past the first postpartum year is associated with higher internalizing, externalizing, and total behavioral problems, further research could assess postpartum maternal mental health longitudinally further into the postpartum period. The literature identifies that perinatal disorders can compromise the quality of parenting and are associated with disturbances in child outcomes [5]. Given the potential for parenting to be a mediator in the relationship between perinatal mental health and child outcomes, further research should consider the inclusion of this variable to understand the complex relationship between C-sections and child behavior further. A limitation is that the sample under-represents women in lower SES categories, as most mothers were university-educated and of high income, affecting the generalizability of the results. There are higher rates of PPD in nations with higher wealth inequality [67]. Although the higher SES sample may be a limitation, the fact that our study identified associations among our sample suggests that lower SES or higher risk groups are likely to demonstrate potentially even stronger concerning associations. The use of maternal self-reports for PTSD, PPD, and CBCL may also be a limitation. The use of the PDSQ for the assessment of postpartum PTSD may not be the most suitable measurement tool and requires further evaluation of its use for the perinatal population. The study controlled for prenatal depression, SES, child sex, and social support; however, given the topic of maternal mental health, future research should consider factors such as prior medication history, a history of intimate partner violence, or the circumstances surrounding pregnancy.

## 6. Conclusions

This study provides insight into the complex factors involved in the association between C-section delivery and child behavioral problems. We found that an emergency C-section was associated with increased PTSD symptoms, which was associated with increased PPD symptoms among new mothers, and in turn, with higher reported child internalizing and externalizing behaviors. Overall, the current study demonstrates the pathway by which an emergency C-section may be associated with child behavioral problems. These findings further support that an emergency operative delivery may harm birth perceptions and thus be associated with altered maternal mental health, putting children at risk of problematic behaviors. Knowledge regarding factors that may impact maternal postpartum mental health and the development of childhood internalizing and externalizing problems will inform prevention and intervention strategies to optimize maternal and child outcomes in the presence of emergency operative delivery.

## Figures and Tables

**Figure 1 behavsci-14-00061-f001:**
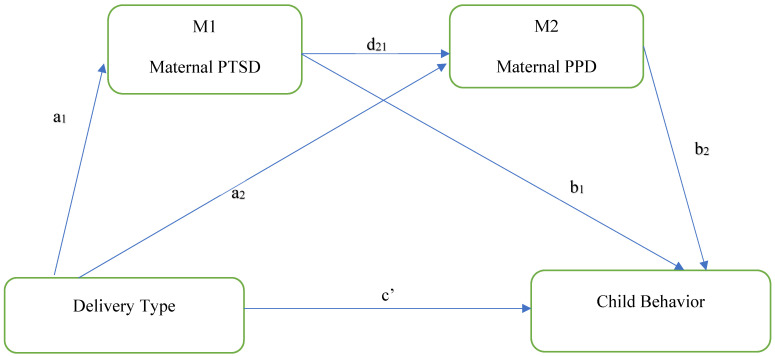
Serial mediation models of the direct and indirect associations between cesarean section type and child behavior. Standardized regression coefficients are a_1_, a_2_, b_1_, b_2_, c′, d_21_.

**Figure 2 behavsci-14-00061-f002:**
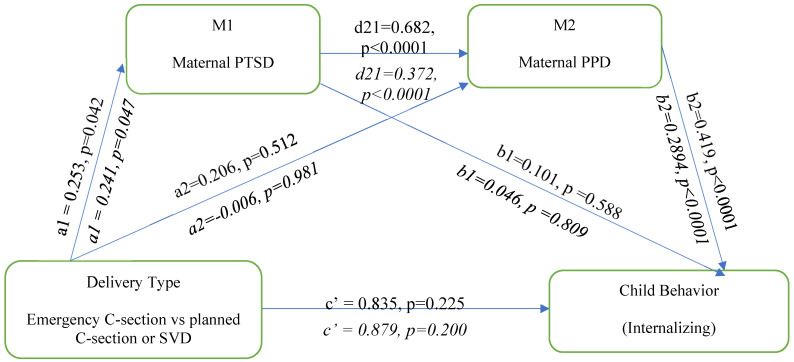
Serial multiple mediation model showing the association between delivery type and child internalizing behaviour; unadjusted and adjusted model. Adjusted model is represented in italics.

**Figure 3 behavsci-14-00061-f003:**
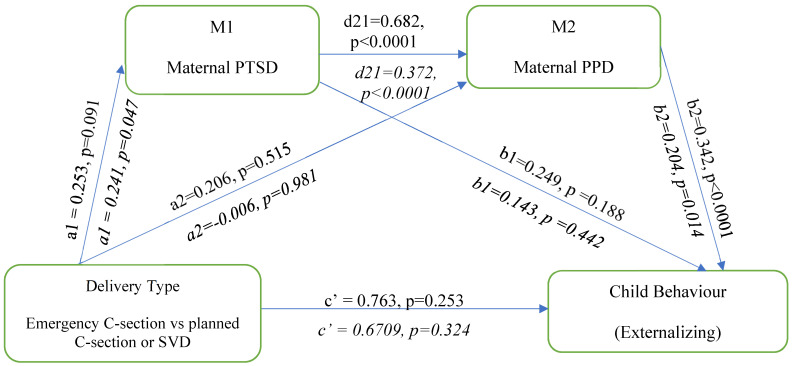
Serial multiple mediation model showing the association between emergency birth and child Externalizing behaviour; unadjusted and adjusted model. Adjusted model is represented in italics.

**Table 1 behavsci-14-00061-t001:** Sociodemographic and descriptive characteristics of study participants n = 938.

Variable	Frequency (n)	Percentage
Maternal education		
Below university degree	227	24.3
University degree or more	705	75.7
Marital status		
Single	24	2.6
Married	914	97.4
Parity		
0	509	54.3
1	330	35.2
2 or more	99	10.6
Household Income		
Less than 70k	147	15.9
70k or more	779	84.1
Delivery Type		
Spontaneous Vaginal Birth/Planned C-section	764	81.5
Unplanned C-section	174	18.6
Gestational age at birth		
<37 weeks	59	6.3
37 or more weeks	879	93.7
Child sex		
Male	487	51.9
Female	451	48.1
PTSD score		
0	702	74.8
1	129	13.8
2	49	5.2
3	32	3.4
4	13	1.4
5 or more	13	1.4
	**Mean (SD)**	**Range**
Maternal age	32.1 (4.2)	17–44
Prenatal depression symptoms	6.6 (4)	0–22
PPD symptoms	4.8 (3.9)	0–20
Prenatal social support	14.8 (2.24)	4–16
PTSD symptoms	0.6 (1.45)	0–11
Internalizing behavior	44.8 (8.3)	29–73
Externalizing behavior	45.8 (8.2)	28–68

**Table 2 behavsci-14-00061-t002:** Indirect effects of emergency birth on child internalizing behavior (unadjusted and adjusted).

Indirect Effect	Effect	Bootstrap SE	95% CI
Total Indirect Effect	0.185 *0.034*	0.164 *0.106*	−0.102, 0.550 *−0.148*, *0.282*
Ind 1 (a): delivery type→maternal PTSD→child internalizing	0.026 *−0.001*	0.065 *0.060*	−0.058, 0.232 *−0.078*, *0.186*
Ind 2 (b): delivery type→maternal PPD→child internalization	0.087 *−0.002*	0.135 *0.080*	−0.171, 0.364 *−0.164*, *0.162*
Ind 3 (c1): delivery type→maternal PTSD→maternal PPD→child internalization	0.072 *0.026*	0.050 *0.023*	0.000, 0.202 *0.000*, *0.096*

Adjusted model is represented in italics. Effect, standardized regression; SE, standard error; CI, confidence interval.

**Table 3 behavsci-14-00061-t003:** Indirect effects of emergency birth on child externalizing behavior (unadjusted and adjusted).

Indirect Effect	Effect	Bootstrap SE	95% CI
Total Indirect Effect	0.192 *0.052*	0.143 *0.087*	−0.054, 0.512 *−0.092*, *0.260*
Ind 1 (a): delivery type→maternal PTSD→child internalization	0.063 *0.035*	0.064 *0.056*	−0.014, 0.269 −*0.031*, *0.216*
Ind 2 (b): delivery type→maternal PPD→child internalization	0.071 *−0.001*	0.011 *0.059*	−0.132, 0.312 *−0.121*, *0.123*
Ind 3 (c1): delivery type→maternal PTSD→maternal PPD→child internalization	0.059 *0.018*	0.041 *0.017*	0.002, 0.173 *0.000*, *0.076*

Adjusted model is represented in italics. Effect, standardized regression; SE. standard error; CI, confidence interval.

## Data Availability

To access these data, please email APrON Principal Investigator, Nicole Letourneau at nicole.letourneau@ucalgary.ca. Data may also be accessed via the PolicyWise Secondary Analysis to Generate Evidence (SAGE) repository (https://policywise.com/sage/ (accessed on 15 January 2021)).

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
