# Peer review of "The Association between Cesarean Section Delivery and Child Behavior: Is It Mediated by Maternal Post-Traumatic Stress Disorder and Maternal Postpartum Depression?"

_behavsci, 2024, doi:10.3390/bs14010061_

Round 1

Reviewer 1 Report

Comments and Suggestions for Authors

Dear authors,

in general I think your article can give an important contribute if could be  included several key aspects, that i will explain, as major revisions.

1- In my opinion your study gives important cues to undestanding the possible influence of delivery mode to child behaviour but it lacks a very important piece of scientific literature that should have been acknowledged in the article. More specifically I think that it is not referrred the different forms parenting behaviour can take over 2/3 years of life and how it can influence the child's behaviour: as examples, it was not controlled if mothers were the main caretakers over all the period, if were present other adverse events in child's life besides maternal mental health issues than can have influenced his/her behaviour. In my opinion it is important to acknowledge that the indirect effect of the delivery mode must be considered in interaction with other variables as parenting capacities and parenting skills. That should be done in introduction when presenting the background for the study, and also in the limitations.

2- Other point that seems very important to me is that authors refer that mothers were included in the study if were present complete data for all the assessment points. However how many were those assessment points? The values included in the study correspond to wich of the points? Or to several of them? If there were several points in time, it would be very important to show the evolution of the measures through time, that would give an important contribution to the strenght of the final conclusions.

3- In section 3.2 it is not described the specific statiscal tests that were used to analyze the data. Authors give the results but do not refer the inferencial statistics used.

Author Response

Reviewer 1

Reviewer Comments

Author Response

 1-In my opinion your study gives important cues to undestanding the possible influence of delivery mode to child behaviour but it lacks a very important piece of scientific literature that should have been acknowledged in the article. More specifically I think that it is not referrred the different forms parenting behaviour can take over 2/3 years of life and how it can influence the child's behaviour: as examples, it was not controlled if mothers were the main caretakers over all the period, if were present other adverse events in child's life besides maternal mental health issues than can have influenced his/her behaviour. In my opinion it is important to acknowledge that the indirect effect of the delivery mode must be considered in interaction with other variables as parenting capacities and parenting skills. That should be done in introduction when presenting the background for the study, and also in the limitations.

Thank you for this comment. I appreciate your comment and agree with the importance of parenting.

There is substantial literature that addresses the role of parenting in child behaviour outcomes. However, including parenting practices as a variable and having a three-mediator model is outside the scope of this paper.

I have addressed the role of parenting in the discussion section in lines 375 to 377.

“Evidence suggests that perinatal disorders, such as maternal PPD, impact parenting quality as symptoms of mental disorders affect an individual’s ability to respond to their environment and thus their parenting capabilities”

Information has also been added to the limitations section on lines 430-435, which now reads: 

“The literature identifies that perinatal disorders can compromise the quality of parenting and are associated with disturbances in child outcomes [5]. Given the potential for parenting to be a mediator in the relationship between perinatal mental health and child outcomes, further research should consider the inclusion of this variable to understand the complex relationship between C-sections and child behavior further.”

2- Other point that seems very important to me is that authors refer that mothers were included in the study if were present complete data for all the assessment points. However how many were those assessment points? The values included in the study correspond to wich of the points? Or to several of them? If there were several points in time, it would be very important to show the evolution of the measures through time, that would give an important contribution to the strenght of the final conclusions

Thank you for this feedback. The section on assessment is provided in sections 2.2.2 and 2.2.3. and indicate the associated time points.

Lines 178-179 now state: “The inclusion criteria were singleton delivery, with complete data on study measures (i.e., EPDS, Psychiatric Diagnostic Screening Questionnaire (PDSQ), and CBCL). “

And lines 213-215:

“Maternal postpartum depressive symptoms were measured at three months postpartum

Maternal PTSD was measured using the PDSQ at three months postpartum.”

3- In section 3.2 it is not described the specific statiscal tests that were used to analyze the data. Authors give the results but do not refer the inferencial statistics used.

Lines 238-240 has been revised to provide more specificity as follows: “To examine whether maternal PTSD and maternal PPD mediated the association between the type of delivery and child behaviour, a linear regression-based serial multiple mediation analysis was conducted using the Hayes process SAS macro version 4.3 (Hayes,2018).”

Reviewer 2 Report

Comments and Suggestions for Authors

This manuscript explores the relationship between having a c-section (planned and emergency) and children’s internalizing and externalizing behaviors in late toddlerhood, and whether this relationship is mediated by maternal mental health (PPD and PTSD). The authors find a link between emergency c-section, maternal mental health, and lower behavior scores for children. They do not find a direct link between c-section delivery and children’s behavior.

This is a well-written, well-organized and careful analysis. I have only a few comments:

1.     This is not a causal analysis. The authors do not say it is, but I suggest they provide some clarity in the strengths/limitations section about the limitations of the research design itself.

2.     The measure of PTSD and PPD is obtained only at one point in time. I believe research suggests that duration of maternal depression might be an important factor in understand impact on children. That is, short periods of depression early on are not the same as chronic depression. Thus, I suggest the authors discuss length of spell and severity of PPD/PTSD as potentially important mediating factors. Perhaps the authors could even test for heterogenous results by PPD/PTSD severity using their measure?

3.     I find the discussion section a bit repetitive, especially of the introduction/lit review.

4.     I find the sample a bit problematic. The authors mention this as a limitation, but a slightly more robust discussion of how their relatively high SES sample might affect the results is important, since those left out from the sample are most as risk of not receiving the support needed.

Author Response

Reviewer 2

Reviewer Comments

Author Response

1. This is not a causal analysis. The authors do not say it is, but I suggest they provide some clarity in the strengths/limitations section about the limitations of the research design itself.

Further information is provided clearly stating that this analysis was not causal. Lines 413 to 417 now reads:

“The use of a multiple linear regression model allowed us to examine the plausibility that maternal mental health is the mechanism through which C-section affects child behaviour [63]. Our study did not aim to identify causation between C-section and child behaviour, but rather, the study aimed to understand how C-section delivery exerts its effect on child behaviour.”

2. The measure of PTSD and PPD is obtained only at one point in time. I believe research suggests that duration of maternal depression might be an important factor in understand impact on children. That is, short periods of depression early on are not the same as chronic depression. Thus, I suggest the authors discuss length of spell and severity of PPD/PTSD as potentially important mediating factors. Perhaps the authors could even test for heterogenous results by PPD/PTSD severity using their measure?

Thank you for this comment. We have further acknowledged the longitudinal patterns of maternal mental health as it relates to child behaviour and presented further information to justify our timeframe as well as future consideration for research. Lines 423-430 now read:

“The study used three-month data for maternal PPD and PTSD as the postpartum onset of PPD is typically diagnosed in the four weeks following delivery Field [64], with the highest onset of PPD and greatest risk of mental health disorder admissions occurring in the first three months following delivery [9, 65, 66]. Given that maternal depression extending past the first postpartum year is associated with higher internalizing, externalizing, and total behavioural problems; further research could assess postpartum maternal mental health longitudinally further into the postpartum period.”

3. I find the discussion section a bit repetitive, especially of the introduction/lit review.

To address the repetitions between the background and discussion, I have deleted small sections of the discussion. Should you feel further revising is required; please provide detailed feedback.

4. I find the sample a bit problematic. The authors mention this as a limitation, but a slightly more robust discussion of how their relatively high SES sample might affect the results is important, since those left out from the sample are most as risk of not receiving the support needed.

A sentence has been added addressing the rates of PPD as it relates to wealth inequality. Lines 435-441 now read:

“There are higher rates of PPD in nations with higher wealth inequality [66]. Although the higher SES sample may be a limitation, the fact that our study identified associations among our sample suggests that lower SES or higher risk groups are likely to demonstrate, potentially even stronger, concerning associations.”

Reviewer 3 Report

Comments and Suggestions for Authors

Intriduction:

Lines 82-83 "Ten to 20 percent of children and youth experience mental health disorders" - worldwide? Please be more specific about this information. Which range of age? Children from 3 to 18?

Lines 98-99: "had higher mean scores on the internalizing (12.05, p<.05) and total problems (48.07, p<.05)" are those values the differences between groups? If not, please present both score values for both groups for a better visualization and interpretation.

Lines 133-134: "Among women with PTSD, 65% also present with PPD, and 22% of women with PPD present with PTSD" please rewrite this sentence. It is difficult to understand which way to interpret this affirmation.

Materials and Methods:

For CBCL - the range of possible scores is missing. Is it from 0 to 200? Please, add this information in the text.

The reference level of significance used in the analysis is missing. Was it 0.05?

Results:

p-values presented as p=0.000: since such a zero value does not really exist for p-values, please adjust the small values to be as p < 0.0001.

Comments on the Quality of English Language

no comments needed

Author Response

Reviewer 3

Reviewer Comments

Author Response

Lines 82-83 "Ten to 20 percent of children and youth experience mental health disorders" - worldwide? Please be more specific about this information. Which range of age? Children from 3 to 18?

Thank you the feedback and enabling me to provide further information. Lines 80-82 have been revised to add the requested detail as follows:

“Ten to 20 percent of children and youth, aged three-to-17 years of age, experience mental health disorders [17, 18].”

Lines 98-99: "had higher mean scores on the internalizing (12.05, p<.05) and total problems (48.07, p<.05)" are those values the differences between groups? If not, please present both score values for both groups for a better visualization and interpretation

As per your suggestion, information has been added to allow for better interpretation. Lines 94-98 now read:

A longitudinal birth cohort (n=172) of mother-infant dyads noted that children born via C-section had higher mean scores on the internalizing (12.05, p<.05) and total problems (48.07, p<.05), as compared to internalizing (9.17) and total problems (39.23) scores of children born vaginally without medications, as measured by the CBCL [4].”

Lines 133-134: "Among women with PTSD, 65% also present with PPD, and 22% of women with PPD present with PTSD" please rewrite this sentence. It is difficult to understand which way to interpret this affirmation.

I have attempted to make the comorbid link between PTSD and PPD clearer. Lines 131-133 now read as “PPD is highly associated with PTSD in the postpartum period. For women with postpartum PTSD, 65% also present with comorbid PPD, and for women presenting with PPD, 22% also present with PTSD [13].”

For CBCL - the range of possible scores is missing. Is it from 0 to 200? Please, add this information in the text.

Line 196 now reads:

“The responses range from 0 to 2, with 0 indicating “not true,” 1 indicating “somewhat or sometimes true,” and two indicating “very true or often true.” The overall score range is from 0-200.”

The reference level of significance used in the analysis is missing. Was it 0.05?

Information added to line 248 “All indirect effects are significant when the p-value is < 0.05”

p-values presented as p=0.000: since such a zero value does not really exist for p-values, please adjust the small values to be as p < 0.0001.

Adjusted throughout text in regards to p value on lines 271, 273, 290, and 292.